# Challenging the Reductionism of "Evidence-Based" Youth Justice

**Stephen Case**

Social and Policy Studies, Loughborough University, Loughborough LE11 3TU, UK; s.case@lboro.ac.uk

**Abstract:** The generation of empirical evidence to explain offending by children and young people has been a central driver of criminological and sociological research for more than two centuries. Across the international field of youth justice, empirical research evidence has become an integral means of complementing and extending the knowledge and understanding of offending offered by the official enquiries and data collection of professional stakeholders and an essential tool for informing 'evidence-based' policy, practice and 'effective intervention'. However, it will be argued that the hegemonic empirical evidence-base created by youth justice research over the past two decades has been generated through methodological reductionism - the oversimplification of complexity, the restriction of conceptual lens and the relative exclusion of competing explanatory paradigms and empirical methodologies, which in turn, has reduced the scope and validity of the policy and practice recommendations derived from it.

**Keywords:** youth justice; empirical; research; evidence; reductionism; risk

The generation of empirical evidence to explain offending by children and young people (hereafter to be framed as "children" in line with the United Nations Convention on the Rights of the Child [1] definition of a child as any individual aged 0–18 years old) has been a central driver of criminological and sociological research for more than two centuries. Youth justice) also known as "juvenile justice" in certain international jurisdictions beyond the United Kingdom) isa concept/label which encapsulates the systems, structures, strategies, processes, organisations and professionals that collectively contribute to the planning and delivery of official responses to offending by children [2]. Across this international field empirical research evidence has become an integral means of complementing and extending the knowledge and understanding of offending offered by the official enquiries and data collection of professional stakeholders and an essential tool for informing "evidence-based" policy, practice and "effective intervention" [3]. Indeed, at the turn of the century, the Chief Social Researcher's Office of the UK Government asserted that "putting the best available evidence from research at the heart of policy development and implementation [enables] well informed decisions about policies, programs and projects" [4]. However, it will be argued here that the hegemonic empirical evidence-base created by youth justice research and its application in policy development processes over the past two decades has been partial in scope, completeness and focus. Tracing the emergence and trajectory of research-informed evidence-based policy and practice into the hegemonic, neo-correctionalist "Risk Paradigm" (animated by the Risk Factor Research evidence-base and preventative practice) will illustrate the overriding methodological reductionism of evidence generation and research utilization in youth justice and England and Wales. This reductionism has manifested in the oversimplification of complexity, the restriction of conceptual lens and the relative exclusion of competing explanatory paradigms and empirical methodologies. In turn this has reduced the scope and validity of policy and practice recommendations and engendered a sustained reliance on "risk" as the central driver of youth justice evidence production.Reductionist evidence generation through youth justice research has been the inevitable by-product of the epistemological dominance of empiricism and positivism over the history of the social sciences, notably within the social science "rendezvous discipline"

of criminology [5,6]. From the birth of empirical explanations of offending in the seminal (statistical) theory of Quetelet [7] criminological theories have privileged the quantitative over the qualitative, the experimental over the appreciative, "hard" over "soft" data and statistics over stories [8]. Positivist criminological research has prioritised quantitative, descriptive knowledge as the key constituent of criminal justice evidence, rather than qualitative, explanatory understanding of offending, in part through the desire to ascribe a "scientific" edifice to their evidence production [5,9] and in part as a response to policy demands for simple, clear research messages to apply in practice contexts [10]. In western-ised youth justice systems, these evidential demands have been manifested in bureaucratic, administrative and technical forms of research characterised by methodological inhibition, partiality (in the dual sense of bias and incompleteness) and fragmented representations of reality ([11–14]). In a study of the most cited scholars in criminology and criminal justice journals in 2015 [15], positivist and experimental criminologists regularly ranked in the top 10 across the highest rated American and international journals. Notably, several of these scholars are youth justice-focused and have epistemological allegiance to the risk factor paradigm [16,17]. Criminologists, academic journals, research assessment panels (for example, the C20 Social Work and Social Policy panel that reviews criminological and criminal justice submissions to the periodic Research Excellence Framework (REF) exercise in England and Wales) and funders of criminological research (for example, the Home Office, Ministry of Justice and ESRC in the UK, the Office for Juvenile Justice and Delinquency Prevention in the USA) have been complicit in consolidating reductionism and partiality by privileging positivist theorising (e.g., the identification of deterministic criminogenic influences), by elevating the status of quantitative, positivist methodologies and statistical significance (as priority over "effect size", for example) to sit atop a hierarchy of evidence generation in criminology [18]. Notwithstanding this reductionism, profes-sional stakeholders in the youth justice field (e.g., politicians, policy makers, practitioners, academics), particularly since the 1990s, have embraced these evidence-bases as valid and comprehensive, introducing two further reductionist, invalidating tendencies into empirical evidence production processes: A "self-fulfilling partiality" (a.k.a. confirmation bias) of consistent replication of reductionist methods and evidence, and a predilection for "synecdoche", where advocates claim that part of a puzzle represents the whole picture when seeking to understand complex issues [9].

## 1. The Emergence of Evidence-Based Policy and Practice in Youth Justice

Towards the end of the 20th century, a period of globalisation swept across the Western world, one animated by rapid socio-economic, political, geographical and technological transformations. The resultant public insecurities, uncertainties and anxieties coalesced into perceptions of these rapid and sweeping transformations as risks and threats needing to be predicted, controlled and managed [19]. A socio-political climate of neo-liberalism emerged, which subsumed concerns with the social contexts of crime within modernising emphases on prescriptions of individual, family and community responsibility, freedom of choice and governance at a distance [20,21]. A concomitant "punitive turn" was evidenced— a movement towards managing the risks presented to the public by allegedly dangerous, threatening children [2] through the increased use of strategies of punishment, control, surveillance and restriction [22]. "Youth offending" isa pejorative label resulting from modernising processes of categorisation and "othering" of populations based on their assessed levels of risk of certain behaviours and the need to control, manage and punish these populations in order to protect the public [22]. In line with the UNCRC-aligned ethos of this paper, the term "offending by children" is preferred. Globalised, neo-liberal and punitive pressures catalysed the modernisation of methods and practices for generating evidence to explain and respond to offending by children by emphasising the role of evidence-based policy as a strategic driver of youth justice responses [23,24].

Evidence-based policy was to be translated into evidence-based practice, a mod-ernising, "scientific" criminal justice approach adopted from the field of medicine that

sought to ensure that all practice was accountable, transparent and defensible. This working model formed part of the rationale for elevating the status of "scientific", quantitative approaches (e.g., Randomised controlled trials (RCTs) across the social sciences. For example, when evidence-based medicine found it impossible to conclusively prove that smoking causes cancer, the field fell back on the notion that smoking was a "risk factor" for cancer—a predictor rather than a cause, but also an explanatory concept that could be readily conflated with causality [10,25]. As such, evidence-based practice (EBP) was introduced and utilised as a tool of managerialism (a neo-liberal strategy of centralised (government) control, management and prescription over the interpretation and implementation of national policy in localised (evidence-based) practice, often animated by regular auditing and monitoring processes, data collection, performance indicators, guidelines, checklists and prescribed procedures [14,26], a guide for practice and resource allocation that signified a move away from the purportedly overly discretionary, less consistent, uncoordinated and expensive systemic responses of the past ([19,27,28]). The application of EBP in youth justice constituted:

> "The conscientious, explicit and judicious use of current best evidence in making and decisions regarding the prevention of offending by individual young people based on skills which allow the evaluation of both personal experience and external evidence in a systematic and objective manner" [29]

## 2. Evidence-Based Practice as Research-Informed

The introduction of EBP served a higher evidential purpose than simply to facilitate modernisation and managerialism of practice, it sought to encourage youth justice professionals to make more use of empirical research evidence (typically from academics) to guide their decision-making [9]. However, from its inception across international youth justice systems, there was a discernible "self-fulfilling partiality" to the development of EBP [30]. For example, the strategies and policies emerging from the Crime and Disorder Act 1998 in England and Wales strongly indicated that "the incoming government already knew what it wanted to do about youth justice" [31]. In this respect, youth justice in England and Wales at this time was reflective of policy development in associated areas. Perhaps most notably, early 21st century drugs policy was founded on a narrowly conceived and reductionist evidence-base [32], whereby attention was paid only to that evidence helpful to the interests of powerful social groups [33], akin to the "Political Model" of research utilization [34] and notions of "policy-based evidence" [35,36].

The newly created Youth Justice Board of England and Wales (a non-departmental public body created to advise government on the development of youth justice policy/strategy and to supporting practitioners with its implementation) was charged with "commissioning research" to inform the development of "effective practice") and consequently "committed to developing and expanding [academic] research . . . to provide evidence that can constructively influence central policy decisions [and] enhance the existent knowledge base" [37,38]. The espoused intention was that academic research take precedence in populating the youth justice evidence-base, as opposed to "less robust" practitioner-generated evidence grounded in knowledge from training, prejudice and opinion, practice experience, anecdote, ads/fashions and advice from senior colleagues [29,36]. However, it soon became clear that the UK Government and its "independent" policy adviser, the Youth Justice Board (YJB), were selectively commissioning and disseminating research that provided evidence to support their preformed neo-liberal policy position on youth justice [39], indicative of a self-fulfilling strategy of generating "policy-based evidence" [35]. Accordingly, the emerging research evidence-base was to explain offending by children and inform youth justice responses in England and Wales, whilst being perpetuated and self-fulfilled by two specific neo-liberal strategies, themselves replicated across other youth justice systems internationally [40]:

- Responsibilisation—assigning primary responsibility to children (also to families and communities) for their own exposure to criminogenic influences, for offending behaviour and for an inability to desist [31];
- Correctionalism—conceptualising and explaining offending by children as the product of identifiable, quantifiable "deficits" and flaws within the individual that can be targeted, treated and allegedly corrected through youth justice sentences and interventions [41,42].

The inter-related neo-liberal strategies of responsibilisation and correctionalism enabled Western governments, working in tandem with a hegemonic group of developmentally minded academic researchers [2], to simultaneously blame children (often disproportionately) for their own exposure to criminogenic influences and to restrict the empirical lens of evidence generation to individualised factors. The corollary of this deliberate reduction of the explanatory evidence-base was the downplaying of the complexity involved in exploring the impact of a broader range of contextual criminogenic factors—structural, political, economic (to compound matters, there is also a history in criminological research and its associated "evidence-based" risk assessment tools of reconstructing and reducing macro influences such as socio-economic deprivation and social marginalisation as individualised risk factors [43,44], cultural, historical, interactional and situational influences [3,23,45] Therefore, a paradox of reductionism began to shape the application of EBP internationally—a necessary, yet potentially invalidating (over) simplification of explanations of offending that offered "an ostensibly neat and coherent approach to the messy and ill-defined complexities of practice" [33].

## 3. Evidence-Based Neo-Correctionalism: The Risk Paradigm

"The management of risk . . . has unified the field [of youth justice], provided significant impetus to attempts to implement evidence-based practice, and generally resulted in a more professional and accountable approach to service delivery" [46]

Across westernised youth justice systems, risk has become the main conceptual lens through which evidence is generated to fulfill neo-liberal responsibilising and correctionalist objectives, with "risk factors" becoming the central "explanatory" concept for the hegemonic, risk-focused youth justice evidence-base. Risk factors constitute quantified representations of problematic and criminogenic experiences, characteristics and "deficits", primarily located in the psychosocial domains of a child's life [47]—the psychological (e.g., emotional, cognitive, attitudinal) and the immediate social (e.g., family, education, neighbourhood/community, peer group). The body of Risk Factor Research (RFR) that forms the evidential basis of these explanations has typically utilised "quantitative scientific methods that can identify potential offenders and reduce recidivism by predicting future behavior" [48]. The data/evidence collection methodologies of RFR often employ binary and scaled measures of offending and exposure to risk [30] such that evidence generation is focused on quantified, aggregated measurements of risk and offending, which are then related through statistical analyses. Accordingly, the explanatory theories that have emerged from the hegemonic quantitative form of RFR [14] have been predominantly developmental, deterministic and neo-positivist—identifying risk factors in early life that are (statistically) predictive (rather than causal) of later offending. The notable exception is constructivist, "pathways" theories that have tended to be qualitative in nature and have focussed on how children actively construct and negotiate their pathways into and out of crime [49,50]. The predilection here is for abstracted empiricism, prioritising psychosocial deficits expressed through probabilistic laws and statistical symbolism that serve to uncritically detach and disembed the individual (child) from their structural influences (e.g., family, neighbourhood, demographic characteristics) of their formations of identity and self [51].

The practical corollary of artefactual RFR evidence is the "Risk Factor Prevention Paradigm" [52,53], which has provided governments with a fit-for-purpose, common sense, modernising and practical approach to youth justice. The Risk Factor Prevention

Paradigm (RFPP) is founded on an evidence-based central preventative premise: "Identify the risk factors for offending and implement prevention methods designed to counteract them" [54]. Taken together, RFR and the RFPP have provided the field of youth justice with an evidenced framework (i.e., "paradigm") and "foundational scientific body of knowledge" [46] for governing the work of practitioners [55] a set of theories, assumptions and ideas about why children offend and what the purpose and content of youth justice professional practice should be (ibid). The evidence-bases of the risk paradigm have proven very attractive to youth justice policy-makers, who have readily accepted the deterministic explanations of artefactual RFR as "universal truths that are stable and reliable" [31]. The RFPP has been embraced as offering "clear, unambiguous guidance on how to solve a problem as complex as offending by children" [31]. However, such evidential certainty has fostered an arguably ill-advised and excessive optimism, even blind faith, amongst youth justice policy-makers regarding the empirical robustness of RFR and the policy-based and practical applicability of the RFPP. This evidential certainty is indicative of what Weiss [38] terms the "Political Model" of social science research utilization in the policy field, whereby research conclusions are privileged if they are "congenial and supportive" in relation to existing political interests/agendas (see also "policy-based evidence"), whilst contrary and new research (evidence) is poorly received and often rejected. As such, evidence-based policy-making should not necessarily be conceived of as linear and to criticise it on this basis would embody the same essentialist and reductionist criticisms being levelled at "evidence-based" youth justice in this article. Nevertheless, as will be discussed, there has been an insidious (sometimes tacit) political framing of evidence-based policy development as linear through the presentation of research utilization as "Knowledge-Driven", implying a direct trajectory from applied research-development-application in practice, and "Problem-Solving", where research provides empirical evidence to solve a policy problem [34] presents other alternative models of research utilization in "evidence-based" policy-making, notably the "Interactive Model" (evidence is sought from a range of stakeholders, including researchers, practitioners, clients and journalists) and the "Enlightenment Model" (research concepts diffuse and influence policy-makers through manifold channels—journals, media, conversations etc). However, neither model reflects evidence-based policy development in youth justice post-1998, which has largely restricted its interactivity to and enlightenment capacity to evidence-based cohering around pre-formed policy/practice agendas [2].

## 4. Evidential Reductionism in the Youth Justice System (YJS) of England and Wales

Post-Crime and Disorder Act 1998, a series of Key Elements of Effective Practice (KEEP) documents outlined the "essential elements of practice with all children at all stages of the YJS" [56] for youth justice practitioners working in England and Wales. It was clear from their outset that the KEEPs would be heavily reductionist, privileging findings from artefactual RFR [31] within their implicit "hierarchy of evidence" [57]. When coordinating the production of the KEEPs and simultaneously guiding government on the development of "effective" youth justice policy and practice, the YJB placed an expectation on KEEP authors to conduct systematic reviews of evidence using the Campbell Collaboration guidelines [58–60]. However, these guidelines are inherently reductionist, elevating experimental and quasi-experimental methodologies (i.e., Randomised Controlled Trials/RCTs) as their gold standard and championing the use of "what works" interventions to prevent crime; interventions with an overriding emphasis on targeting psychosocial risk factors [61–63]. Indeed, the predominance of artefactual RFR evidence across the KEEP documents rendered "certain research question "unaskable" because they cannot be addressed using experimental methods" [58]; thus negating any potential for the production of "inconvenient evidence" [34,41]; concurrently depersonalising and deprofessionalising the recommended practice of youth justice staff through a prescribed adherence to the risk lens [26].

The central, pivotal KEEP was "Assessment, Planning Interventions and Supervision" (APIS), which outlined the "foundation activities which guide and shape all work with young people who offend" [56]. APIS prescribed "a consistent risk management methodology resting on a platform of knowledge" [29], with the mooted benefits for enabling objective, standardised and evidence-based (risk) assessment previously not possible through clinical, discretionary models [64,65]. The "dependable methods" prescribed by APIS were primarily restricted/reduced to the application of the newly created "Asset" structured risk assessment instrument [66], which offered a "rigorous evidence-based assessment" [56]. The guidance for achieving rigorous assessment cautioned practitioners against the reductionism of "relying on a favourite or fashionable theory" [56] when explaining offending, yet the same guidance counterintuitively dictated that assessments be informed by a restricted group of developmental, artefactual RFR theories: Criminal Careers [53], the Age-Graded Informal Social Control Theory [18] and Interactional Theory [17]. Consequently, the Asset assessment instrument generated an evidence-base through practice that was overwhelmingly populated by the "risk factors associated with offending behaviour" [56] that had been widely replicated in artefactual RFR and which were all situated within psychosocial risk categories/domains (living arrangements, family and personal relationships, education/training/employment, neighbourhood, lifestyle, substance use, physical health, emotional/mental health, perception of self and others, thinking and behaviour, attitudes to offending, motivation to change). Thus associated planning, judgements and decisions were framed almost entirely and inevitably by risk evidence and associated explanations. Practitioners were instructed to assess exposure to risk factors as a binary measure (yes/no) and to quantify their perceptions of the extent to which exposure to risks aggregated across each domain were associated with "the likelihood of further offending": From 0 (no association) to 4 (very strong, clear, direct association). Quantitative judgements were supplemented with qualitative, narrative explanations in a small, summative "evidence box" at the end of each section [30].

There is clear, thorough-going conceptual, methodological and practical reductionism to the risk-led youth justice of the Crime and Disorder Act 1998 in England and Wales; processes that have mobilised over two decades of "evidence-based" practice shaped by "effective practice" guidance and assessment tools. The central reductionist act of oversimplifying complex and dynamic aspects of children's lives (e.g., their experiences, interactions, perceptions, thoughts) into readily quantifiable "dynamic" (i.e., targetable, malleable) and restricted groups of psychosocial "risk factors" could never hope to represent the "lived realities" of those children [67]. Nor could it encompass the full range of criminogenic influences in children's lives, which may include psychosocial factors, but also "needs, motives, knowledge, social deficits … [and] social and physical contextual factors" [3]. Youth justice research evidence generation has been rendered even more reductionist in explanatory terms by the oversimplified outcome measures of risk assessment tools (e.g., binary measures of reoffending and risk exposure), with limited sensitivity to offence seriousness, frequency, duration etc or consideration of alternative (non-risk) measures of effectiveness (e.g., increases to positive outcomes). Risk assessment in the YJS of England and Wales has embodied a staged process of reductionism that has rendered risk more of a generalised and dehumanised artefact than a practical explanatory concept [68–71]. The economic and practical sustainability of Asset risk assessment can be attributed to the self-fulfilling reductionism of evidence generation in research and practice, which has perpetuated a restricted psychosocial evidence-base by exclusively employing a risk lens to explain offending by children.

Reductionist evidence generation in the YJS in England and Wales peaked in November 2009 with the inception of the "Scaled Approach" assessment and intervention framework, which dictated that every child subject to court disposals was to have the level of their intervention (frequency, intensity and nature) matched/scaled to their Asset score: Low risk/standard intervention, medium/enhanced, high/intensive [72,73]. The Scaled Approach effectively formalised evidential reductionism in the YJS by consolidating risk (fac-

tors) as the primary conceptual and explanatory animator of youth justice policy and practice that was largely bereft of a robust theoretical foundation or philosophical/principled core. However, the framework has been criticised for its uncritical use of aggregation, which "inevitably imposes limits on the accuracy" of these predictions [74], which reduces understanding of the risk profiles and life experiences of individual children (the "ecological fallacy") and which potentially invalidates any proposed intervention [75]. Intervention validity was further reduced by the Scaled Approach's inherent partiality—privileging individualised, psychosocial interventions as responses to assessed psychosocial risk factors (the focus of Asset assessment), so "attention is drawn away from structural, social inequalities for which government itself has some responsibility" [30]. Consequently, the Scaled Approach was criticised on the same grounds of practical (un)sustainability as its central Asset component, indeed, extending processes of risk-based reductionism and invalidity into the sphere of intervention.

## 5. Reductionist Risk Reliance and Reduced Explanations

"Despite the publication of thousands of studies and chapters referring to dynamic risk factors ... virtually none of them acknowledge their problematic theoretical status" [3]

The deterministic and partial nature of the hegemonic, risk-informed evidence-base generated by contemporary youth justice practice in England and Wales exemplifies and perpetuates the reductionism of youth justice research. Privileging and replicating artefactual RFR evidence of "dynamic" (malleable) risk factors to populate explanations of and responses to offending by children has been asserted to increase the robustness and utility of the youth justice evidence-base. Yet this has inevitably reduced the scope, completeness and validity of the explanations provided by this same evidence-base, most notably in relation to:

- Mechanisms—in RFR, explanatory mechanisms are understood and conceptualised as neo-positivist, statistical "predictors" rather than "causes", thus prioritising prediction over theory [76]. Additionally, the factorisation of risk has produced simple, comprehensible and practical research evidence for application in practice. However, the greater the extent of factorisation, the greater the (over) simplification of the lived real-life experiences of children and the less accurate and representative of real life the data becomes. In efforts to try to explain more, the explanatory theories of RFR actually explain less—artificially reducing understandings of offending by children. Furthermore, oversimplistic factorisation processes deliberately neglect any examination of causality [54], which may be harder to measure and evidence due to their complex, multi-faceted and qualitative nature. Nonetheless, predictive risk factors are often presented as deterministic and implicitly/assumptively (not evidentially) understood as causal;
- Breadth—RFR has perpetuated a decontextualised and responsibilising focus on individualised, psychosocial domains of life when exploring and purportedly identifying the origins of offending, rather than examining potential contextualised criminogenic influences such as situations, relationships, interactions, system contact, demographic characteristics, individual constructions of personal experiences and a multiplicity of economic, political and socio-structural factors [45,61,77]. This decontextualisation has been exacerbated in the past decade by the (re)emerging North American evidence-base of biological (quasi) positivism promoting the predictive and criminogenic influence of "low resting heart rate" as a key risk factor for offending, with the associated causal mechanisms identified as impulsive sensation seeking, fearlessness and the need for physiological arousal [78–80];
- Validity—when practice employs the RFPP, children are filtered and conceptualised through actuarial social categories or (latterly) emerging algorithmic data patterns [77], rather than explanations informed by individual, constructed identities [49]. Therefore, the risk-focused explanations of offending behaviour that underpin and animate the

RFPP have been aggregated across and attributed to groups based on their collective risk category or level, rather than on the risk profile of individual children. The methodological/practical side effect is ecological fallacy, whereby unsupported and potentially invalid statistical inferences are made about the nature of individual children based on the (risk) profile of the group to which they belong. Additionally, representing individuals through aggregated risk categories may well facilitate better estimates of risk of reoffending, but may also do little to aid understanding of the causes of this offending and thus to guide interventions and treatments [3];

- Definity—advocates of RFR have confidently disseminated definite, clear-cut, evidence-based "explanations" predicated on indefinite, unspecific and inconsistent definitions and understandings of central concepts [30,74]. There remains ambiguity and lack of empirical consensus, even within the artefactual RFR community, over the nature of "risk factors" an explanatory concept (e.g., variously understood as causes, predictors, indicators, correlates, symptoms) and their relationship with offending (e.g., deterministic or probabilistic? causal or predictive? linear or concurrent?). At best, this indefiniteness should reduce policy and practice confidence in the validity and applicability of the artefactual RFR evidence-base. At worst, it could totally invalidate the conclusions of artefactual RFR, especially if risk factors cannot be conclusively linked with offending beyond their identification as statistical correlates, which would mean that they are not even predictive of offending and thus should not be understood as "risk factors" at all.

The foregoing debates indicate that every stage of the (predominantly) risk-led production of the youth justice research evidence-base has been driven by reductionist partiality, privileging neo-positivist, developmental, artefactual risk factor theories [24] and explanatory mechanisms (i.e., predictive, psychosocial risk factors) over alternative explanatory mechanisms, paradigms and evidence-bases. These reductionist biases have fostered further partiality (incompleteness) in the nature and scope of potential explanatory theories and mechanisms that populate the "effective practice" evidence-base (e.g., the content of risk assessment inventories). Therefore, descriptive knowledge has taken precedence over explanatory understanding of offending by children, yet this knowledge has been represented as indicative of a complete and unequivocal evidence-base for youth justice practice [9].

## 6. The Expansion of Reductionism: Resilient Risk Reliance

Faced with sustained and mounting criticisms of the paradoxically reductionist, yet expansionist (e.g., interventionist, net-widening, expensive) nature of "evidence-based", risk-led youth justice in a climate of growing economic austerity (emerging circa 2010), the YJB developed the "AssetPlus" assessment and intervention framework to replace the Scaled Approach. AssetPlus ostensibly offers a more comprehensive, contextualised and dynamic assessment method of explaining offending and thus informing interventions that is founded in non-risk/positivist, qualitative evidence-bases and evidence-generation processes that solicit children's voices, promote positive behaviours and explore contextual criminogenic influences [81]. These changes reflect a nascent culture shift and paradigm shift in youth justice evidence production in England and Wales, away from reductionist, deficit-based risk approaches to understanding and responding to offending by children and towards holistic, prospective, appreciative and optimistic understandings of children's lives [82,83]. Further to this, YOT practitioners have received training in desistance theory [84] and the Good Lives Model [85]—both of which were developed with adult offenders but both of which enable greater focus on the individual's strengths and positive behaviours.

However, a reductionist reality soon emerged following the inception of AssetPlus circa 2015. While AssetPlus was introduced with the rationale that evidence would not be quantified, practitioners were required to provide a summary prediction of the "indicative" and "final" likelihood of reoffending (low, medium, high) using a three-point quantified

scale reminiscent of the Scaled Approach risk categories [27] signifying a worrying evidential retreat into risk-led reductionism. Indeed, thus far in practice implementation, AssetPlus has been unable to break free from the conceptual and methodological shackles of the risk paradigm [82,83]. Whilst the framework promises potential improvement on the deleterious reductionism of old, it does not constitute a comprehensive overhaul of youth justice evidence production and has yet to depart substantively in practice from the longstanding neo-liberal, neo-correctionalist risk focus [83]. Similarly, practice guidance materials and practitioner feedback have been indicative of a resilient (reductionist) reliance on risk over and above the proposed emphasis on strengths and the future-orientated promotion of positive behaviours and outcomes [49,82]. Consequently, for many practitioners, particularly the majority who have been "hard schooled" in risk-led evidence generation throughout their training (with very limited re-training regarding AssetPlus), the new framework has been mobilised using a risk lens as a default mechanism to help staff make sense of an ambiguous, exhaustive framework by reverting to "business as usual" [82].

More broadly, a reliance on risk-based research and reductionist methods of collecting evidence in youth justice practice has persisted across international justice systems, most notably across the YJS of England and Wales. The Ministry of Justice (MoJ) is the UK Government department in charge of managing and monitoring the Criminal Justice System, which includes the YJS designated for children aged 10–17 years old. A guidance document published by the MoJ, "What works in managing young people who offend? A summary of international evidence" [86], concluded that practice should place primary importance on "assessing the likelihood of further offending" (i.e., risk prediction), prioritising "attributes that are predictive of reoffending" (i.e., risk factors) and "matching service to that level of risk" (ibid: 1), equivalent to the reductionist "Scaled Approach". The subsequent "Prevention in youth justice" briefing report from the YJB, part of their "Effective practice in youth justice" series, summarised "what works" in preventing offending by children as those approaches integrating the "strong evidence about risk factors for offending—a mix of personal, environmental and social factors" [87]. Despite the prevention briefing presenting practice evidence of the effectiveness of diversionary interventions and programmes which prioritise positive activities, its reductionist risk reliance foregrounded these discussions and was elevated to the top of the implicit hierarchy of evidence within the document. In direct contradiction of the progressive recommendations of the contemporaneous review of the YJS [88], the MoJ and YJB reports offered support for continued adherence to generating artefactual RFR/RFPP evidence when pursuing youth justice objectives, from the prevention of first-time offending [87] to the reduction of reoffending by identified offenders [86]. Repeated iterations of Her Majesty's Inspectorate of Probation criteria for inspecting YOTs also focus on collecting evidence of how well YOTs are "mitigating and responding to identified risks" [89], albeit with an additional, broader emphasis on the assessment of strengths, wider societal factors and structural barriers [89]. The persistent and resistant focus on reducing offending and exposure to risk factors introduces yet more reductionism into youth justice evidence-generation—the conceptualisation of "effectiveness" as simplistic binary measures of offending and risk.

### 7. Trojan Horses of Risk Reliance

A paradox of "increased reductionism" is discernible across contemporary Western youth justice systems, signified by the rise to prominence of the Algorithmic Risk Assessment and the Adverse Childhood Experiences movements as "evidence-based" shapers of youth justice policy and practice. Crucially, both developments have the potential to be conceptualised and animated as reductionist and actuarial "Trojan Horses" for youth justice based on their overriding reliance on employing the risk lens to identify, conceptualise, measure and target criminogenic influences.

Algorithmic Risk Assessment is the new kid on the block for generating empirical and explanatory evidence through assessment mechanisms and the use of algorithms (finite sequences of well-defined, computer-implementable instructions and data patterns). This

encapsulates a new episteme for knowledge production in youth justice—a conception of human nature radically different from the rational and pathological (positivist) epistemes that have previously dominated evidence generation in youth justice. As such, Algorithmic Risk Assessment (ARA) ostensibly represents an evolution (although some would argue devolution) of risk assessment methodologies beyond actuarial logics [90]. ARA proffers a performative, a-theoretical, predictive and non-reflexive conception of offending by children that is entirely lacking any humanistic component, privileging surface knowledge over depth of understanding [77]. While Asset risk assessment was actuarial, identifying predictive risk factors based on established, replicated research evidence from artefactual RFR, ARA eschews scientific, empirical foundations, trusting computational algorithms to identify patterns in the data. Algorithmic forms of assessment purport to increase the accuracy and efficiency of risk prevention by offering complete precision, devoid of the subjective bias (partiality) of actuarial risk assessment. This is due to collecting data (as explanatory "evidence") from diverse sources, rather than gathering restricted psychosocial data specifically for the purposes of risk prediction [91]. Through complex algorithmic analysis, patterns of data subsume and supersede social categories, with individuals filtered and conceptualised through the patterns that emerge inductively from their data [77] rather than the actuarial preference for identifying predictors deductively from replicated RFR evidence.

Despite its burgeoning popularity across westernised youth justice systems, ARA is extremely limited in its ability to generate evidence to inform youth justice interventions, notably because its deliberate lack of a research foundation. In ARA, "explanations" of offending are situated within an evidential "black box", rendering it impossible and unnecessary to explain how risk scores have been deduced (unlike in actuarial assessment) and so reducing (to the point of elimination) any consideration of the aetiology of offending by children [92,93]. Indeed, in this respect, ARA arguably returns evidence generation to its criminological roots in the superficiality of Quetelet's seminal identification statistical correlates as explanations (predictors) of offending patterns in societies [7]. Through ARA, there is a greatly reduced (if any) necessity for empirical research in youth justice or any requirement to collect research evidence relating to human beings, demographic groups or social situations [91,93]. ARA and the practice decision-making that it informs are not intended only to be empirically defensible or "evidence-based" [92], a stark contrast to and arguably a devolution from the transparent and accountable (albeit empirically partial) actuarial, risk-led foundations of EBP. Thus, ARA signifies a move away from generating research "evidence" to inform practice in youth justice, yet also signifies a potential exacerbation of the overly technical, prescribed and non-reflective "understandings" and decision-making engendered by actuarial risk assessment.

Furthermore, although ARA allegedly circumvents the pitfalls of partiality inherent in actuarial methods (e.g., due to its explicit lack of pre-formed evidence-base), it has been asserted that "it actually smuggles in all sorts of biases, assumptions, and drivers of inequality" [94]. It is highly likely that ARA suffers from "layers of bias" [95], for example, disproportionately and artefactually identifying black and minority ethnic groups as posing higher risk (of reoffending, of harming others) by implicitly privileging the psychosocial risk domains/factors during assessment, which groups are more likely to be exposed to and disadvantaged by. Accordingly, the identification of minority groups as "high risk" through ARA can compound existing socio-structural biases and disadvantages and also compound their existing over-representation in the YJS [65,96]. As such, ARA functions as a Trojan Horse for risk-based reductionism in the youth justice field by dehumanising and decontextualising "evidence" of criminogenic influences, with potentially deleterious effects on the recipients of youth justice interventions.

Another potential Trojan Horse for continued reliance on risk-led research and practice evidence is the Adverse Childhood Experiences (ACEs) movement—the fastest growing empirical research evidence-base in contemporary youth justice. At the time of writing (November 2020), increasing prominence is being assigned to conceptualising evidence

from ACEs research as explanatory in the YJS of England and Wales [97] and internationally, especially in the USA [98,99]. The burgeoning ACEs research evidence-base conceptualises traumatic experiences in childhood (e.g., psychological, physical and sexual abuse, physical and emotional neglect, household dysfunction, mental illness, substance abuse, divorce) as predictive of harmful behaviours and negative outcomes in later life, including offending, aggression, imprisonment, psychiatric disorders, school exclusion, substance use and poverty [100–102]. ACEs exposure is often measured quantitatively using a cumulative risk approach that involves totalling the number of adversities experienced by a child in order to evaluate and predict their "risk" of negative outcomes [103]. Consequently, the common-sense, practical and familiar (risk-led) nature of the ACEs evidence-base has proven readily amenable to application in the youth justice field. This ostensible utility, for example, in relation to children with multiple complex needs, has encouraged swift acceptance across the youth justice policy-making and practitioner community. The logical practice corollary has been to advocate for integrated (multi-systemic) responses to offending that utilise ACEs as their central explanatory mechanism through a focus on health, wellbeing, psychology and behaviour [104] (Taylor 2019).

The emerging research evidence-base for ACEs-focused explanations of, and responses to offending by children requires close scrutiny, particularly when its accelerating popularity is set against the continued risk reliance of youth justice systems. There is a distinct danger across international youth justice systems that ACEs are being conceptualised and swiftly adopted by governments and policy-makers as psychosocial "quick fixes" to more complex social issues [105] or worse still, as an over-simplistic and stigmatising proxy for continued reliance on explaining offending by children using psychosocial risk factors [106]. For example, when examining the use of ACEs in youth justice interventions, researchers have employed the category/lens of "Risk Factors" to conclude that "there are factors, warning signs and issues that can alert us to potentially problematic behavior" [100]. Additionally, ACEs are being increasingly conceptualised and measured in actuarial ways, with children rated above a certain cumulative threshold of ACEs considered to be at higher "risk" of negative outcomes and thus meriting more intensive intervention (i.e., akin to the "Scaled Approach"). Both quick fix and proxy understandings portray the ACEs evidence-base as a potential Trojan Horse for continued reductionism in youth justice.

Further to this potential, thus far, there has been an inordinate degree of imputation from policy-makers, service providers and academic stakeholders [100,104,107] regarding the causal (deterministic) and predictive influence of ACEs on negative behaviours and outcomes in later life, particularly the uncritical attribution of aggregated/population risk levels to individuals [106]. This purported (i.e., imputed) causality and predictive validity serves as the rationale for retrospective practitioner assessments of ACEs, often leading to a quantified ACEs "score" to guide intervention (i.e., the "factorisation" of ACEs). However, the ACEs evidence-base, particularly in terms of youth justice research, is not yet sufficiently developed to offer conclusive evidence of the predictive validity of ACEs themselves or explanations of their relationship with offending—indicative of a conceptual and empirical ambiguity equivalent to that demonstrated by artefactual RFR. Despite ongoing evidential ambiguity and partiality (e.g., psychosocial bias—equivalent to that of artefactual RFR), in its current form, an unjustified (non-evidenced) degree of policy pre-emption has followed the nascent ACEs evidence-base. The YJB Strategic Plan [97], for example, has committed to rolling out explicit, "trauma-informed" responses to ACEs exposure across the YJS England and Wales, whilst sustained and convincing evaluation results of the efficacy of such an approach are absent. It should be noted that trauma-informed practice can have a much broader lens than indicated by the current dominance of the psychosocial, risk-focused ACEs agenda, particularly the acknowledgement that children who offend are likely to have experienced greater levels of trauma, which need to be addressed when working with them [106]. This traumatic background does not have to be quantified or conceptualised in deterministic and predictive ways as it is within ACEs research and practice.

This pre-emptive approach is reminiscent of the "policy-based evidence" agenda of the Labour Government that introduced the Crime and Disorder Act 1998 [35], where a self-fulfilling evidence-base was generated to validate preferred policy through post-hoc research and consultation with practice [73] (e.g., validation of the Scaled Approach using problematic and inconsistent evaluation results). Therefore, there is an imminent danger that governments and policy-makers may (continue to) co-opt, repurpose, distort and exploit the ACEs agenda in the service of continued risk reliance.

## 8. Reducing Reductionism: Expanding the Evidence-Base

There has been intransigent and resilient risk reliance across youth justice systems internationally (notably in England and Wales) in recent years, which has illustrated and perpetuated evidential reductionism. However, continued economic pressures and sustained challenges to the reductionist risk paradigm have motivated some key stakeholders in academia (not least the contributors to this special issue) and policy/practice fields to broaden their evidential purview in relation to alternative, less reductionist/more expansionist empirical, evidence-bases considered of utility within (and beyond) the youth justice field. This contemporary explanatory open mindedness (i.e., expansionism) has not been necessarily reflexive or principled in origin or motive in all instances, nor could it hope to be in the politically charged and economically driven context of youth justice [108]. Ironically, a central driver of expansionist theoretical, empirical and practice evidence generation in youth justice has been the dramatic reduction of resources in the area and the pragmatism that this has catalysed.

Austerity-driven service retrenchment has simultaneously discouraged centralised prescriptions from policy-makers in respect of the nature of evidence generation and its application in practice, whilst encouraging decentralised, localised practice innovations, mediations and discretion. This has precipitated a degree of tentative expansionist innovation at the local level [109,110]. Allied to this expansionism, strategic developments in youth justice and related policy fields (e.g., health, education, social care, policing) at the inter/national level have moved evidence generation in youth justice (slightly and occasionally) away from its reliance on applying artefactual RFR evidence animated through the RFPP (e.g., the Scaled Approach). Academics, policy-makers and practitioners generating evidence to explain offending by children have been given the space and permission (although rarely the centralised funding) to pursue new approaches to examining this complex issue from different angles, old and new.

This changing context of evidence production has precipitated a renewal of interest in the empirical evidence-bases of historically popular aetiological/theoretical movements in youth justice ("the old") inter alia, diversion and systems management [111,112], social control/bonds theory [113] and restorative justice [114] (Restorative Justice Council 2017). There has been a concurrent push to develop new evidence-bases around the modern, progressive and anti-risk/positivist approaches ("the new") such as those focused on constructive resettlement (constructive (e.g., strengths-based, positive), coordinated (e.g., multi-agency, multi-sector), consistent (e.g., stable), customised (e.g., individualised) and co-created (e.g., participatory, inclusive) responses to children who have offended [115] following contact with the YJS [101,115]), relationship-based practice and engagement methods [116], trauma-informed working [115] and the development of a "Child First" strategic model of Positive Youth Justice [75,117]. These evidence-bases are being developed and applied through a combination of application (e.g., repurposing established evidence-bases in the contemporary youth justice context), modernisation (e.g., updating or reframing existing evidence-bases) and innovation (e.g., creating new evidence-bases through new ways of thinking). Each strategy has necessitated rejecting, challenging or deemphasising the methodologies, findings and conclusions of RFR in order for the YJS to address the inherent reductionism and synecdoche of past evidence generation in youth justice.

A prime example of the expanding explanatory purview of youth justice evidence generation in England and Wales is the emergence of the "Child First" national strategic objective. In 2016, a wholesale review of the YJS of England and Wales concluded that youth justice should generate more evidence focusing on and incorporating broader "child first" and education-focused understandings of offending by children, rather than more traditional deficit-led, risk-based explanations [88]. These recommendations were to incorporate contextualised, economic and socio-structural explanatory evidence, thus eschewing traditional psychosocial, decontextualised risk bias and explicitly encouraging evidence generation across a broader range of life domains (e.g., welfare, education, social care) that address the "multiple complex needs" of children who offend [88,118]. The review recommendations have been reflected in subsequent strategic developments in England and Wales, with youth justice policy, strategy and practice standards now underpinned (at least rhetorically) by the official systemic objective of "Child First"—responding to children who offend primarily in terms of their child status and by addressing their best interests, needs, rights, strengths and capacities in constructive, collaborative and non-criminalising ways [98,113,115,117]. In this way, the YJB and other key stakeholders in England and Wales are supportive of and amenable to the generation of a broader evidence-base to explain offending by children; one that would have been previously viewed as "inconvenient" by politicians and policy-makers relying on risk and reductionism to pursue "policy-based evidence" agendas fuelled by actuarialism, individualisation, penal expansionism and populist punitiveness [41].

## 9. Conclusions: The Bifurcation and Cognitive Dissonance of Youth Justice Research Evidence

An "irony of sustainability" has become entrenched within youth justice research in the past three decades. Modernising, neo-liberal pressures for sustainable approaches to policy and practice development that are "evidence-based" have encouraged stakeholders to enhance the role and importance of empirical research, which has been manifested in the privileging of risk evidence and mobilised by a "Political Model" of research utilization [34]. The persistence of the risk-led reductionism of "evidence-based" policy and practice in youth justice requires scrutiny. This is particularly so in a contemporary strategic climate that encourages cognitive dissonance by necessitating sensitive contextualisations of offending by children alongside a continued reliance on risk-led conceptualisations and explanations founded in aggregation and generalisation. Why is such cognitive dissonance persistently encouraged? Why is one hand of youth justice stimulated to generate new and improved evidence-bases that are fit-for-purpose in addressing the complexity of children's lives, whilst the other hand is compelled to perpetuate the self-fulfilling prophecy of risk reliance, underpinning evidence-based research, policy/strategy, practice and training exclusively with RFR evidence? For example, the training course leading to accreditation of youth justice staff via the "Foundation degree in youth justice" and the "Youth justice effective practice certificate" [119] includes a "Risk" module reflecting the Scaled Approach to assessment and intervention. However, other available models reflect newer and more expansionist evidence generation in youth justice around topics such as desistance, relationship-based practice, engagement and wellbeing.

As a starting point to addressing these complex questions, can lessons be learnt from the thorough-going critique of reductionist evidence generation presented in this article? It has been argued that the hegemonic empirical evidence-base underpinning policy development has been partial (i.e., biased, incomplete), restricting conceptual lenses, explanatory paradigms and empirical methodologies to those privileged by quantified (artefactual) positivist and quasi-positivist "risk factor" approaches, thus rendering resultant policy and practice recommendations similarly partial and narrowly framed. In England and Wales (also internationally), sustained risk reliance has combined with economic austerity and been exacerbated by the restrictions of the ongoing COVID-19 crisis to foster a climate of bifurcation and cognitive dissonance in contemporary youth justice, wherein continued investment (practically, financially, occupationally, systemically) in risk-based youth justice

sits uneasily alongside practical expansionism and local innovation. Of particular concern is the potential for stakeholders to attempt to reconcile this cognitive dissonance through an uneasy, messy and contradictory combination of improved evidence generation and sustained risk reliance [101,115].

The origins of reductionist evidence generation in the YJS are clear, as are the self-fulfilling and self-validating processes that have maintained and expanded its dominance. However, a new question emerges for contemporary youth justice: why reductionist risk reliance persists in the face of sustained critiques and broader and more developed evidence-bases regarding the aetiology of offending by children and appropriate youth justice responses to it? One uncomfortable answer lies in the continued impotence of criminological critique and critics (including myself) to exert significant impact on evidence-based policy development in youth justice (notwithstanding more recent green shoots of optimism within the "Child First" agenda), akin to the limited impact of "public criminology" and academic criminologists more broadly [51,120]. Another (ironic) answer lies in sustainability, particularly sustained investment in the reductionist risk management agenda.

Fundamentally, many youth justice stakeholders (including academics) have invested heavily in the risk agenda over a number of years and at a number of levels: Politically (e.g., supporting specific political imperatives) [121], economically/financially (e.g., developing risk assessment inventories, conducting RFR), systemically (e.g., prioritising risk-led processes of assessment, intervention and inspection), conceptually (e.g., constructing, validating and applying RFR theories and evidence), strategically (e.g., producing risk-informed national policy and practice guidance) and practically (e.g., training, educating and "hard schooling" staff in risk management approaches and their associated evidence-bases) [24,119]. High levels of investment have inevitably precipitated (even demanded) a partiality for particular forms and uses of "evidence" (i.e., from artefactual RFR) to sustain, self-fulfil and validate, often at the expense of generating alternative evidence that could serve as a "complicating inconvenience" politically, economically, strategically and practically [121]. This partiality has understandably produced an occupational culture of risk-led practice in the YJS, exemplified by the "business as usual" default to a risk approach adopted by practitioners trying to make sense of the new AssetPlus framework [82]. Moreover, more than two decades of investment in the risk paradigm has fostered resistance to alternative approaches to conceptualising and responding to offending by children (e.g., diversionary, strengths-focused, "Child First"). This is at least partly due to their underdeveloped evidence-bases and a lack of training, guidance, time and money to reinvest in non-risk-led methodologies—artefacts of the self-fulfilling prophecy of risk reliance.

The empirical evidence-base derived from youth justice research has been subject to an incrementally "increasing reductionism" over time, with the hegemonic conceptual, methodological and practical lens consistently narrowing. For example, from the determinism of positivism to the predictive utility of RFR and latterly factorised, individualised ACEs evidence; from rational and pathological epistemes to dehumanised and decontextualised algorithms. These mechanisms of iterative reductionism, further shaped and reduced by conceptual and practical partiality, continue to artificially/artefactually restrict the validity of the empirical evidence generated to explain the offending by children and, by extension, to limit its applicability and sustainability in practice. Therefore, many of the problems and limitations inherent to evidence generation in the youth justice field have been, and continue to be artefacts of "how research is done and used, rather than what is being researched" [9]. Contemporaneously, there has been pressure exerted by youth justice stakeholders for academic research to identify explanatory "universal truths" and practical "quick fixes", which may have encouraged some research stakeholders to over-claim regarding the explanatory comprehensiveness of empirical evidence [9] and to be overly optimistic regarding the potential of research evidence to shape, inform and influence policy development [23].

Thus, excessive extrapolation and optimism derived from research are themselves products of reductionism—constant pressures to produce simplistic and digestible explanations of a highly complex phenomenon—offending by children. The self-fulfilling partiality and the persistence and resilience of risk reductionism in conceptual, methodological and practical spheres of youth justice evidence generation have yet to be fully acknowledged or addressed by key stakeholders. It is hoped that a dynamic combination of sustained critique of reductionist risk reliance and the emerging context of practical expansionism, local innovation and "progressive" conceptualisations of children who offend (e.g., Child First objectives, trauma-informed practice as non-deterministic and strengths-based) will fill the vacuum of evidence generation in westernised youth justice, producing valid, holistic, fit-for-purpose and child-friendly alternatives to delivering youth justice in the 21st century.

**Funding:** This research received no external funding.

**Informed Consent Statement:** Not applicable.

**Data Availability Statement:** Not applicable.

**Conflicts of Interest:** The authors declare no conflict of interest.

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
