# Peer review of "Challenging the Reductionism of “Evidence-Based” Youth Justice"

_sustainability, doi:10.3390/su13041735_

Round 1

Reviewer 1 Report

This is a detailed review article bringing together the literature on evidence-based policy making (broadly defined) with the field of youth justice. The discussion centres around the narrow understanding of the possibilities of evidence that have been generated by various inter-related processes. The author illustrates that these entail the ne0-liberal socio-economic climate of the late twentieth century, which occurred alongside a wide-ranging movement towards risk management and control within social policy under the aegis of what Beck (1992) terms the risk society. A net outcome of these developments is that understandings of youth justice have consolidated in recent years around notions of risk and propensity. That is around individuals propensity to engage in risky behaviours over which they have little power to resist (responsibilisation) , but within which they are also blamed for their susceptibility and thus need to be controlled/punished (correctionalism). The article goes onto argue how the framing of youth justice in this way has lent itself to particular forms of enquiry which have prioritised quantitative methods as being the most appropriate methods to demonstrate these mechanisms.  

There is a clear narrative that runs through the piece. It provides a clear overview of the trajectory of Youth Justice Policy and practice from the late twentieth century into the twenty-first century. It demonstrates the continuity that has happened alongside and even within the changes. See the discussion on ACEs and Trauma-Informed approaches.

The conclusion makes some hard-hitting points about the entrenchment of the risk paradigm within policy, practice and beyond. Some are heavily invested in the policy, but is it equally the case that careers have been made as active critics, but that there has been limitations to this critique as the system is still firmly entrenched despite the adverse evaluations of its efficacy? These might be worthy of further reflection. If the article concentrates on the limitations of evidence, might it not also be the case that those arguing from a critical persuasion have also experienced a successful failure akin to the broader discipline of criminology (see Loader and Sparks 2011).

I would have no hesitation in recommending this article for publication. I have reserved a couple of points worthy of elaboration for the discussion on the nature of evidence-based policy making.

First, there have been long-standing warnings of the dangers of abstracted empiricism in the social sciences for many years. Not least in the work of C Wright Mills (1959) but these themes have been developed further in the work of Jock Young (2004) and his critique of voodoo criminology – it strikes me that this is worthy of comment. It also links to the point raised above about the lack of impact of critique

Second, there is an accruing literature that shows how debates around the use of evidence are nuanced. This can be traced back to the pioneering work of thinkers such as Carol Weiss who introduced different models of evidence utilisation. The point is that the tendency to base criticisms of evidence-based policy making on the limitations of the linear version (see Weiss, 1986) where evidence (primarily generated via natural scientific methods) has a direct impact on policy are somewhat guilty of essentialism themselves. This is not because they are inaccurate per se, but because there is a whole literature that has emerged to show the alternative ways in which evidence can impact on policy. This article would be enhanced by consideration of some of that work ( see Weiss, 1979; Nutley et al, 2007)

On page Page 3 Lines 79-90 – the author suggests that the government already knew what it wanted to do in terms of YJ Policy and that the evidence was, in effect, cherry picked to support this. Again highlighting that YJ was in this respect no different to other policy areas might be worthy of comment (see the various analyses of UK Drug Policy Making at this time for a clear point of comparison (e.g. Stevens, 2007). There is a further level to this and it links to the longer trajectory of how evidence was conceived narrowly. The author might consider here the statements by the then Secretary of State for Education, David Blunkett’s analysis of evidence-based policy in a speech delivered to the ESRC – See Monaghan (2011). Although signalling the New Labour Government’s commitment to move beyond the ideology and dogma of the Conservative Governments of Thatcher and Major which the succeeded, it has been argued that the narrow appreciation of evidence-based policy merely replaced one dogma with another (possibly a political dogma for a methodological one)

Page 5 Line 174 – What are the Unaskable questions? This seems like a point worthy of brief elaboration – the subsequent point also fits in with some of the critiques made of evidence-based policy under the modernising agenda; i.e. that Governments are often quite contemptuous of evidence as it makes the process of arriving at decisions much more arduous. We are seeing this play out in real time with the barring of appointees to Advisory Councils for what are seemingly innocuous reasons https://www.theguardian.com/politics/2020/feb/19/drugs-advisory-panel-candidate-was-blocked-after-criticism-of-jeremy-hunt

Suggested references:

Loader, I., & Sparks, R. (2013). Public criminology?. Routledge.

Nutley, S. M., Walter, I., & Davies, H. T. (2007). Using evidence: How research can inform public services. Policy press.

Monaghan, M. (2011). Evidence versus politics: Exploiting research in UK drug policy making?. Policy Press.

Stevens, A. (2007). Survival of the ideas that fit: an evolutionary analogy for the use of evidence in policy. Social Policy and Society6(1), 25-35.

Weiss, C. H. (1979). The many meanings of research utilization. Public administration review39(5), 426-431.

Young, J. (2016). Voodoo criminology and the numbers game. In Cultural criminology unleashed (pp. 27-42). Routledge-Cavendish.

Author Response

Each point is addressed in red type in the attached document

Reviewer 2 Report

This article addresses an important area in relation to recent trends in youth justice in the UK.

I am familiar with the tools and approaches which are discussed in the article, and can confirm that the critical appraisal of  these have real applicability to  current policy and practice in this field. The discussion and analysis is also of  wider interest in the criminal justice, social welfare and social work ‘risk’ debates. Some interesting points are made  re the over determining nature of some tools- which has had an effect on views on ‘evidence’ about,  and possibly  attempts of ‘prediction’ of,  offending /reoffending.

It is very useful in its account  of the  development of ‘risk’ ideas and models, and provides a very good analysis  points of interest re current debates. It provides an excellent and detailed review/analysis of recent influences on  societal views/ government agency  policies.

At times, it could be more succinct, and I think that there could be an introduction which more readily  prepares the reader for the main features and progression of argument within the article, with a rounding  off of this in a  clearer way towards the end of it. With those points in mind, I would recommend  minor amendments in relation to the presentation and progression of argumentation   points rather than anything substantial within the article itself, such as  a few more paragraphs  re key learning points at the end concerning   the previously signposted influences arising from the discussion/debates re the complexity  of risk factors, the limits of positivism and reductionism  per se.

Author Response

(The authors gave the same response as above.)
